# A Two-Stage Particle Swarm Optimization Algorithm for Wireless Sensor Nodes Localization in Concave Regions

**Yinghui Meng \*, Qianying Zhi, Qiuwen Zhang and Ni Yao**

School of Computer and Communication Engineering, Zhengzhou University of Light Industry, Zhengzhou 450002, China; 331907040397@zzuli.edu.cn (Q.Z.); 2012032@zzuli.edu.cn (Q.Z.); yaoni@zzuli.edu.cn (N.Y.)

**\*** Correspondence: yinghuimeng@zzuli.edu.cn; Tel.: +86-136-3386-3376

**Abstract:** At present, range-free localization algorithm is the mainstream of node localization method, which has made tremendous achievements. However, there are few algorithms that can be used in concave regions, and the existing algorithms have defects such as hop distance error, excessive time complexity and so on. To solve these problems, this paper proposes a two-stage PSO (Particle Swarm Optimization) algorithm for wireless sensor nodes localization in "concave regions". In the first stage, it proposes a method of distance measuring based on similar path search and intersection ratio, and completes the initial localization of unknown nodes based on maximum likelihood estimation. In the second stage, the improved PSO algorithm is used to optimize the initial localization results in the previous stage. The experimental result shows that the localization error of this algorithm is always within 10% and the execution time is maintained at about 20 s when the communication radius and beacon node ratio is changing. Therefore, the algorithm can obtain high localization accuracy in wireless sensor network with "concave regions", requiring low computing power for nodes, and energy consumption. Given this, it can greatly extend the service life of sensor nodes.

**Keywords:** nodes localization; concave region; intersection ratio; similar path; particle swarm optimization algorithm

## 1. Introduction

### 1.1. Research Significance

The WSN (Wireless Sensor Network) is a distributed sensor network, and its tip is a sensor node that can perceive physical, chemical, behavioral, and biological information in the external environment. The nodes in WSN communicate through wireless, so the deployment of the network is simple, the setting is flexible, and it can be connected to the Internet through wireless [1]. Therefore, this technology has been widely used. When the sensor node in WSN is working, it sends the physical, chemical, behavioral, and biological information collected from the environment to the gathering node, then the gathering node transmits it to the internet or terminal computer. In the end, users analyze the obtained information and carry out corresponding operations. However, in actual application, it is useful only when the information collected by sensor nodes is combined with the coordinates. For example: in a large farmland where WSN is deployed, a certain node has monitored the drought situation of crops in its location. If the user also knows the coordinates of this node while receiving the information, then the user only needs to accurately irrigate the crops there. Otherwise, the information transfer by the node will be valueless. Therefore, the sensor node localization technology is the core technology of WSN.

At present, only GPS localization and manual configuration can obtain the exact location of nodes. However, those methods are unsuitable for WSN due to the high cost and limited usage environment. In recent years, many researchers have conducted in-depth research on WSN node localization technology, and have achieved rich scientific research results. Nowadays, there are two categories of node localization algorithms which the range-free and range-based.

Range-based localization algorithms require sophisticated ranging instruments and ideal communication conditions. In practical applications, adding additional ranging equipment to nodes is expensive, and the propagation of signal is extremely vulnerable to the environment and weather conditions. Therefore, the range-based localization algorithm cannot obtain qualified localization results in low-cost and great-scale wireless sensor networks.

Range-free localization algorithms do not require the support of additional ranging technology. The coordinates of target nodes can be calculated based on the connection information among nodes. With the advantages of energy saving and high computing efficiency, this way is more applicable to the large-scale wireless sensor networks with complex deployment environment. Classic range-free methods include DV-hop localization algorithm, centroid algorithm and so on.

If there exist a communication area *N*, take the line between any two points in *N*. If any point on the line segment is in area *N*, then the area is convex; otherwise, the area is concave. In Figure 1, (a), (b) are concave regions, (c), (d) are convex regions. WSNs are mostly deployed in inaccessible environments for human, such as mountains, lakes and swamps. Under these environmental conditions, large obstacles are prone to appear in the communication area of the sensor network, which will cause the communication area to become concave. Therefore, under the condition that the communication area is a concave area, it is the core technology of WSN to calculate the location of unknown nodes accurately by using range-free location algorithms.

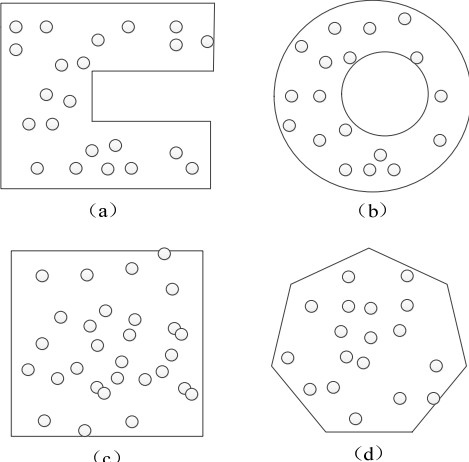

**Figure 1.** Comparison diagram of concave and convex areas. (**a,b**) are concave regions; (**c,d**) are convex regions.

## 1.2. Research Status

The current wireless sensor nodes localization methods for concave areas could be classified into two categories.

The first category: According to structural properties of the wireless sensor networks and the coordinates of some beacon nodes, unknown nodes are located by methods such as beacon node selection, communication area division, and shortest distance correction.

Li Mo et al. proposed a range-free Rendered Path (REP) localization algorithm for concave regions in Reference [2]. First, the algorithm assumed that the boundary nodes deployed around the obstacle are known, then determined the shape and location of the network cavity. Based on the location of obstacle, the shortest paths between all nodes in the network is classified into several

sub-segments, then, the Euclidean distance between nodes is calculated by constructing a virtual unit circle at the intersection of the concave boundary and the shortest path between nodes. Although this method has improved the estimation accuracy of Euclidean distance between nodes, the main disadvantage is that the communication volume and energy consumption of the algorithm are very huge, and the computing power of the node is relatively high. Lim et al. proposed a Proximity-Distance Map (PDM) algorithm in Reference [3], in this algorithm, the estimated Euclidean distance and real Euclidean distance between all beacons nodes in WSN are represented by matrices, then, the estimated distance are processed according to the linear transformation matrix of the two matrices, so as to improve the estimation accuracy of Euclidean distance. A location calculation method was proposed in Reference [4], which uses cubic spline interpolation to filter out the shortest path affected by the concave boundary, improving the estimation accuracy of the distance. According to friendly beacon nodes selection, Paul et al. proposed a Friendly Anchor Based Range Free Localization (FABL) algorithm in [5]. The algorithm used the true distance and estimated distance between beacon nodes to calculate an angle value, the unknown node takes the eight beacon nodes with the largest angle value for localization calculation. Bulusu et al. proposed a Convex-Hull Partitioning (CHP) localization algorithm in [6]. The CHP algorithm divided the beacon nodes into convex hulls. After the partitioning, the shortest distance between beacon nodes in each convex hull are uninfluenced by the concave shape, then the coordinates of the target unknown nodes are calculated based on the convex hull where they are located.

The second category of localization methods use intelligent optimization algorithm to locate unknown nodes.

Z. Zhang et al. improved the particle swarm algorithm in [7] and proposed the waves function, which uses waves function to represent the performance of particles location. Aiming at the shortcomings of DV-Hop (Distance Vector-Hop), in Reference [8], Ahmad et al. came up with a coordinate estimation method based on artificial bee colony algorithm, which reduces the localization error by restricting the optimized area. Mirjalili et al. proposed an improved Gray Wolf Optimization (GWO) localization algorithm in Reference [9], this algorithm simulates the social hierarchy of wolves dividing unknown nodes into four levels: $\alpha$, $\beta$, $\gamma$ and $\omega$, guiding $\omega$ to complete localization with $\alpha$, $\beta$ and $\gamma$.

On the basis of the structure of the sensor network, the first category of localization algorithms reduces the influence of the concave boundary on the shortest distance between nodes through methods such as beacon node selection, shortest distance correction and so on. However, when estimating the Euclidean distance between nodes, it still uses the DV-HOP algorithm to calculate the Euclidean distances, which will cause a larger hop distance error, thereby greatly reducing the localization accuracy of the nodes. In Figure 2, $O$, $A$, $B$, $C$ are sensor nodes, and $R$ is the radius of the nodes which are in the WSN. The distances of $OA$, $OB$, $OC$ are obviously different. $O$ to $A$, $B$, and $C$ are all one hop, if the hop number and hop distance are used to estimate the distance of $OA$, $OB$, $OC$, $OA = OB = OC$ is obtained, so the calculation result is inconsistent with the actual situation, this is the hop distance error. The first category of localization algorithm uses the estimated distance with hop distance error to calculate the location of the nodes, which will inevitably cause a larger localization error. The second category of localization algorithms directly introduces intelligent optimization algorithms to locate unknown nodes through thousands of iterations, which reduces the overall localization error of some target nodes. However, due to lack of constraints, the coordinates of individual nodes even may be located outside the network. In addition, thousands of iterations require huge energy consumption, which will greatly shorten the service period of sensor nodes. In summary, in view of the shortcomings of the above localization algorithms, in this paper, a two-stage PSO algorithm for wireless sensor node localization in the concave region is proposed. The first stage: base on the similar path and intersecting ratio to determine whether the multi-hop shortest path between nodes is affected by the concave boundary, then calculate the distances between target unknown nodes and beacon nodes, finally, using the least square method to complete nodes localization. The second stage: using the improved PSO method to optimize the coordinates which are calculated in the previous stage. Experiment results

indicate that this method uses intersection ratio to estimate the distance between nodes, so it can effectively avoid the hop error. In addition, using the improved PSO to optimize the consequence of the initial stage could greatly reduce the number of iterations and extend the service life of nodes.

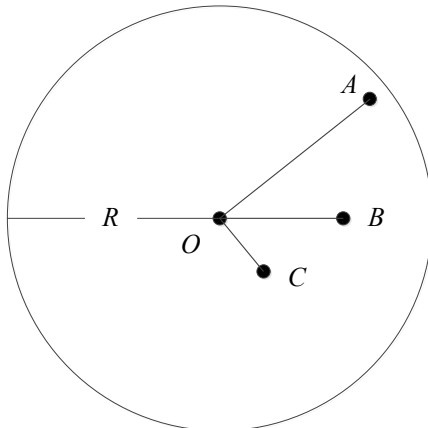

**Figure 2.** Schematic diagram of hop error.

## 2. Initial Localization of Unknown Nodes

### 2.1. Calculate the Euclidean Distance of One Hop Based on the Intersection Ratio

In WSNs, the target unknown node could be located only after the Euclidean distance between nodes in the communication area is obtained. First of all, this paper uses the method of intersection ratio to estimate the Euclidean distance of one hop. Then the Euclidean distances between the target unknown node to beacon nodes are calculated basing on the multi-hop shortest path between nodes and the Euclidean distance of one hop. In this paper the intersection ratio is: the ratio of the distance between nodes to the radius.

As shown in Figure 3, $a$, $u$ are the nodes with communication radius $R$, and they are neighbor nodes with each other. $S_{au}$ is the area of the intersection part, $N_a$ is the amount of sensor nodes in the circle of the node $a$, $N_u$ is the amount of sensor nodes in the circle of node $u$, the amount of sensor nodes in the intersecting area is $N_{au}$, the Euclidean distance between $a$ and $u$ is $d_{au}$, the intersection ratio between nodes $a$ and $u$ is $\frac{d_{au}}{R}$. So the area $S_{au}$ of the intersection is:

$$S_{au} = 2R^2 arccos(\frac{d_{au}}{2R}) - d_{au}\sqrt{R^2 - \frac{d_{au}^2}{4}} \tag{1}$$

Divide by $\pi R^2$ to get:

$$\frac{S_{au}}{\pi R^2} = \frac{2}{\pi}\arccos\left(\frac{d_{au}}{2R}\right) - \frac{d_{au}}{\pi R}\sqrt{1 - (\frac{d_{au}}{2R})^2} \tag{2}$$

There is only one unknown number $\frac{d_{au}}{R}$ on the right side of Formula (2), and the intersection ratio $\frac{d_{au}}{R}$ can indicate the distance between neighbor nodes, so assign $x = \frac{d_{au}}{R}$ to get:

$$\frac{S_{au}}{\pi R^2} = \frac{2}{\pi}\arccos\left(\frac{1}{2}x\right) - \frac{x}{\pi}\sqrt{1 - \frac{1}{4}x^2} \tag{3}$$

According to Taylor Expansion:

$$\begin{cases} \arccos(x) = \frac{\pi}{2} - x - \frac{1}{6}x^3 - \frac{3}{40}x^5 - \cdots\cdots \\ x\sqrt{1-x^2} = x - \frac{1}{2}x^3 - \frac{5}{8}x^5 - \cdots\cdots \end{cases} \tag{4}$$

Substitute Formula (4) into Formula (3):

$$\frac{S_{au}}{\pi R^2} = \frac{1}{\pi}\left(\pi - 2x + \frac{1}{12}x^3 + \cdots\cdots\right) \tag{5}$$

The value range of $x$ is $[0, 1]$, so, the polynomials after cubes can be ignored:

$$x \approx \frac{\pi}{2}\left(1 - \frac{S_{au}}{\pi R^2}\right) \tag{6}$$

Which is:

$$\frac{d_{au}}{R} \approx \frac{\pi}{2}\left(1 - \frac{S_{au}}{\pi R^2}\right) \tag{7}$$

In the communication circle $u$ or communication circle $a$, the probability of a sensor node located in the intersection area and outside the intersection area approximately satisfies the Poisson distribution model [10]. Its probability function is:

$$P = \frac{(\lambda D)^k}{k!}e^{-\lambda D} \ (k = 0, 1, \cdots\cdots n) \tag{8}$$

The ratio of the intersection area to the communication circle area is $\frac{S_{au}}{\pi R^2}$, the ratio of the amount of nodes in intersecting area to the amount of nodes which are in the communication circle is $\frac{2N_{au}}{N_a + N_u}$, The values of the two ratios should be approximately equal, so:

$$\frac{S_{au}}{\pi R^2} = \frac{2N_{au}}{N_a + N_u} \tag{9}$$

Substituting Formula (9) into Formula (7), the intersection ratio is:

$$\frac{d_{au}}{R} = \frac{\pi}{2}\left(1 - \frac{2N_{au}}{N_a + N_u}\right) \tag{10}$$

The estimated distance of a hop between node $u$ and node $b$ is:

$$d_{au} = \frac{\pi R}{2}\left(1 - \frac{2N_{au}}{N_a + N_u}\right) \tag{11}$$

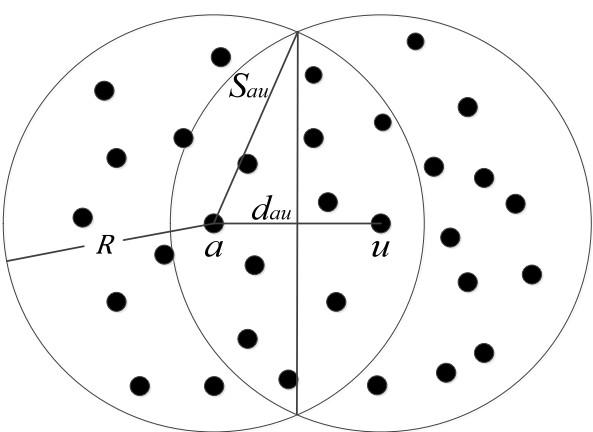

**Figure 3.** Schematic diagram of calculating the distance between neighbor nodes.

## 2.2. Judge Whether the Multi-Hop Shortest Path between Beacon Nodes Is Affected by Concave Boundary

In the concave region, there are two cases when calculating the Euclidean distance between nodes: whether the multi-hop shortest path between sensor nodes influenced by the concave boundary.

So, this paper proposes a method for judging whether the shortest multiple paths between nodes are affected by the concave boundary, and a similar path search algorithm to calculate the Euclidean distance between nodes.

### 2.2.1. Judgment Method

As shown in Figure 4, $u_i$ are target unknown sensor nodes, $a_1, a_2, a_3, a_4$ are beacon nodes, the solid line is the multi-hop shortest path, the dashed line is the actual shortest path. Affected by the concave boundary, the multi-hop shortest path between nodes $a_3$ and $a_4$ is quite different from the actual shortest path. Since the multi-hop shortest path between nodes $a_1$ to $a_2$ is not affected by the concave boundary, it is similar to the actual shortest path. Therefore, when estimating the Euclidean distance between $a_1$ and $u_3$, the method $d_{a_1 u_3} = d_{a_1 u_1} + d_{u_1 u_2} + d_{u_2 u_3}$ can be used. If using the same method to estimate the Euclidean distance between $a_3$ and $u_4$ by adding the distance of each hop on the multi-hop shortest path, it will inevitably result in a great error [11]. Therefore, in the concave region, there are two cases when calculating the Euclidean distance between nodes: whether the multi-hop shortest path between sensor nodes influenced by the concave boundary.

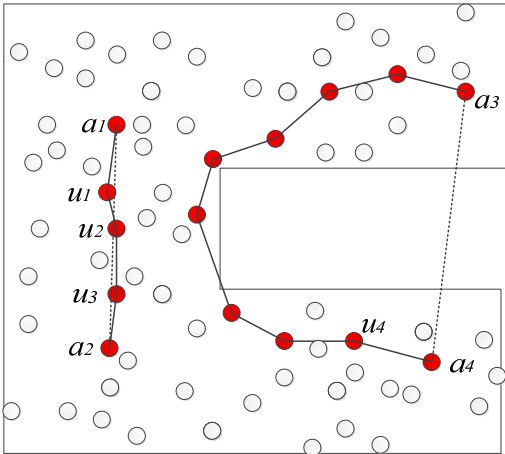

**Figure 4.** Schematic diagram of the multi-hop shortest path between nodes.

This paper proposes a similar path search algorithm, the purpose is to find a multi-hop shortest path between beacon node pair, which is most similar to the multi-hop shortest path of the target to beacon node. After judgment, if the path is affected by the concave boundary, the multi-hop shortest path that beacon node to the target node is also affected by the boundary; Otherwise, it is not affected. For example: in Figure 4, when judging whether the multi-hop shortest path that from $u_4$ to $a_3$ is affected by the concave boundary, first, according to the similar path search algorithm, it can be obtained that the multi-hop path between the beacon nodes $a_3$ and $a_4$ is most similar to the multi-hop path between $u_4$ and $a_3$. Then, it is judged that the multi-hop path between $a_3$ and $a_4$ is affected by the concave boundary, so, the multi-hop shortest path between $u_4$ and $a_3$ is also affected by the concave boundary.

The method for judging whether the multi-hop shortest path between beacon nodes is affected by the concave boundary is as follows:

1. Calculate the distance $d_{a_i a_j}$ between beacon nodes $a_i$ and $a_j$. Divide $d_{a_i a_j}$ by the communication radius $R$ to obtain the true shortest path intersection ratio between beacon nodes $\frac{d_{a_i a_j}}{R}$.

2. According to Formula (11), figure out the distance of each hop. The distance of each hop on the multi-hop shortest path between beacon nodes is summed to obtain the multi-hop path

distance $d_{MHD}$ (Multi-Hop Distance), and then divided by the communication radius $R$, Obtain the intersection ratio $\frac{d_{MHD}}{R}$ of the multi-hop shortest path between beacon nodes. If there is:

$$\left| \frac{d_{MHD}}{R} - \frac{d_{a_i a_j}}{R} \right| \approx 0 \tag{12}$$

it is determined that the multi-hop shortest path between beacon nodes is not affected by the concave boundary; otherwise, it is affected by the concave boundary.

### 2.2.2. Similar Path Search Algorithm

This paper proposed a similar path search algorithm, which aims to looks for a multi-hop shortest path between two beacon nodes. It requires that the path is most similar to the multi-hop shortest path between unknown node and the target beacon node. The algorithm uses the *Ochiai* coefficient [12] to measure the similarity between paths. The larger the value of *Ochiai*, the higher the similarity between paths; the smaller the value of *Ochiai*, the lower the similarity between paths. The calculation method of *Ochiai* coefficient is:

$$Ochiai(A, B) = \frac{N(A \cap B)}{\sqrt{N(A) \times N(B)}} \tag{13}$$

$A$ and $B$ are sets, $N(x)$ is the amount of elements contained in the set $x$. The details of the similar path search algorithm are:

1. Find the multi-hop shortest path of the target unknown node to $a_i$. Record this sensor nodes passed by the path into set $A$.
2. Find the multi-hop shortest paths from the remaining beacon nodes which are in the WSN to node $a_i$, and record the nodes passed by these paths into the sets $B_1, B_2 \cdots\cdots B_j \cdots\cdots B_n (i \neq j)$.
3. Calculate the *Ochiai* coefficients of $A$ and $B_1, B_2 \cdots\cdots B_j \cdots\cdots B_n$ respectively.
4. Arrange the *Ochiai* coefficient values obtained in step 3 in descending order. If the coefficient value corresponding to $Ochiai(A, B_j)$ is the largest, the multi-hop shortest path from $a_i$ to $a_j$ is taken as the most similar path Most Similar Path (MSP) of the target unknown node to $a_i$.

For example, as shown in Figure 5, $a_1, a_2, a_3, a_4, a_5, a_6$ are beacon nodes, $u_1, u_2, u_3, u_4, u_5, u_6, u_7, u_8, u_9$ are unknown nodes. When looking for the most similar path that $u_1$ to $a_6$, first, find the multi-hop shortest path from $u_1$ to $a_6$, and record the sensor nodes on the path into the set $A$, so set A is: $\{u_1, a_2, u_3, u_6, u_7, a_6,\}$. Then record the multi-hop shortest path that between the remaining beacon nodes and $a_6$, and record the nodes on these paths to the set $B_1, B_2 \cdots\cdots B_j \cdots\cdots B_n (i \neq j)$. The results are as follows: $B_1 = \{a_1, u_2, u_3, u_6, u_7, a_6\}$, $B_2 = \{a_2, u_3, u_6, u_7, a_6\}$, $B_3 = \{a_3, u_4, u_3, u_6, u_7, a_6\}$, $B_4 = \{a_4, u_5, u_4, u_3, u_6, u_7, a_6\}$, $B_5 = \{a_5, u_6, u_7, a_6\}$.

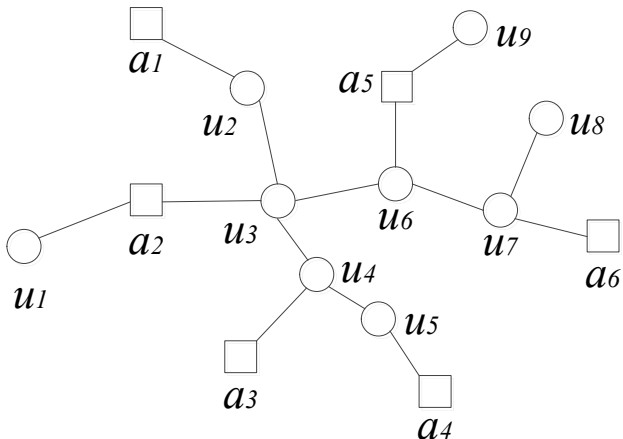

**Figure 5.** Schematic diagram of similar path search.

According to Formula (11), the *Ochiai* coefficients of path $A$ and other paths can be calculated as: 0.667, 0.913, 0.667, 0.617 and 0.612. Therefore, the multi-hop shortest path between $a_2$ and $a_6$ is used as the MSP from $u_1$ to $a_6$.

### 2.3. Calculate Distance and Estimate Location

#### 2.3.1. Calculate Distance

Base on the above, in the concave region, there are two situations in which the multi-hop shortest path is affected by the concave boundary and not affected by the concave boundary. The distance estimation method in these two cases is as follows:

(1) When the multi-hop shortest path is not affected by the concave boundary, according to Formula (11), the sum of the distances of each hop between neighbor nodes on the multi-hop shortest path is used as the estimated distance between nodes. For example, as shown in Figure 6, $u_i$ is unknown nodes, $a_i$ is beacon nodes. If the multi-hop shortest path between $u_3$ and $a_1$ is not affected by the concave boundary, so, the distance between $u_3$ and $a_1$ is:

$$d_{a_1 u_3} = d_{a_1 u_1} + d_{u_1 u_2} + d_{u_2 u_3} \tag{14}$$

The values of $d_{a_1 u_1}, d_{u_1 u_2}, d_{u_2 u_3}$ can be obtained according to Formula (11).

(2) The multi-hop shortest path is affected by the concave boundary. As shown in Figure 6, if the distance from $u_4$ to $a_3$ and the distance from $u_5$ to $a_3$ need to be calculated, the multi-hop shortest path between beacon nodes $a_3$ and $a_4$ is known, and this path is the MSP of $u_4$ to $a_3$ and $u_5$ to $a_3$. So $u_5$ is on the MSP, $u_4$ is outside the MSP. Therefore, when the multi-hop shortest path is affected by the concave boundary, there are two situations that the unknown node is on the MSP and the unknown node is outside the MSP.

When calculating the distance from $u_5$ to $a_3$, it is known that the unknown node $u_5$ is on the MSP. As shown in Figure 6, connecting $a_3$, $u_5$, draw a perpendicular line from $u_5$ to $a_3 a_4$ in triangle $a_3 a_4 u_5$, and the perpendicular line intersect $a_3 a_4$ at point $g$, according to the Pythagorean theorem:

$$d_{u_5 a_3} = \sqrt{d_{u_5 a_4}^2 - d_{a_4 g}^2 + (d_{a_3 a_4} - d_{a_4 g})^2} \tag{15}$$

The value of $d_{u_5 a_4}$ in Formula (15) can be calculated according to Formula (11), so there is only one unknown number $d_{a_4 g}$ in Formula (15). The distance between $a_4$ and $g$ can be estimated according to the intersection ratio of paths and the *Ochiai* coefficient of MSP. The calculation method is as follows:

$$d_{a_4 g} = d_{u_5 a_4} \times \frac{d_{a_i a_j} / R}{d_{MHP} / R} \times Ochiai_{u_5 a_3} \tag{16}$$

In Formula (16), $Ochiai_{u_5 a_3}$ is the *Ochiai* coefficient of the multi-hop shortest path between nodes $u_5$ and $a_3$. Substituting Formula (16) into Formula (15):

$$d_{u_5 a_3} = \sqrt{d_{u_5 a_4}^2 + d_{a_3 a_4}^2 \times \left(1 - \frac{2 \times d_{u_5 a_4} \times Ochiai_{u_5 a_3}}{d_{MHP}}\right)} \tag{17}$$

When the distance between $u_4$ and $a_3$ needs to be calculated, and $u_4$ is outside the MSP, similarly:

$$d_{u_4 a_3} = \sqrt{d_{u_4 a_4}^2 + d_{a_3 a_4}^2 \times \left(1 + \frac{2 \times d_{u_4 a_4} \times Ochiai_{u_4 a_3}}{d_{MHP}}\right)} \tag{18}$$

In summary, when the $u_k$ is on the MSP, the estimation formula of the distance from $u_k$ to $a_i$ is:

$$d_{a_i u_k} = \sqrt{d_{DSP}^2 + d_{a_i a_j}^2 \times \left(1 - \frac{2 \times d_{DSP} \times Ochiai_{u_k a_i}}{d_{MHP}}\right)} \tag{19}$$

When the unknown node $u_k$ is outside the MSP, the estimation formula of the distance from $u_k$ to $a_i$ is:

$$d_{a_i u_k} = \sqrt{d_{DSP}^2 + d_{a_i a_j}^2 \times \left(1 + \frac{2 \times d_{DSP} \times Ochiai_{u_k a_i}}{d_{MHP}}\right)} \tag{20}$$

the distance from $u_k$ to $a_i$ is $d_{a_i u_k}$; the multi-hop shortest path between beacon nodes $a_i$ and $a_j$ is the MSP of $u_k$ to $a_i$, and the distance from $a_i$ to $a_j$ is $d_{a_i a_j}$; The multi-hop path between $u_k$ and $a_i$ has a non-coincidence part with its MSP. The non-coincidence path is defined as Dissimilar Path (DSP), and $d_{DSP}$ is the length of the DSP; $Ochiai_{u_k a_i}$ is the *Ochiai* coefficient that the multi-hop shortest path from $u_k$ to $a_i$; $d_{a_i a_j}/R$ is the actual shortest path intersection ratio of $a_i$ and $a_j$; $d_{MHP}/R$ is the multi-hop shortest path intersection ratio of $a_i$ and $a_j$.

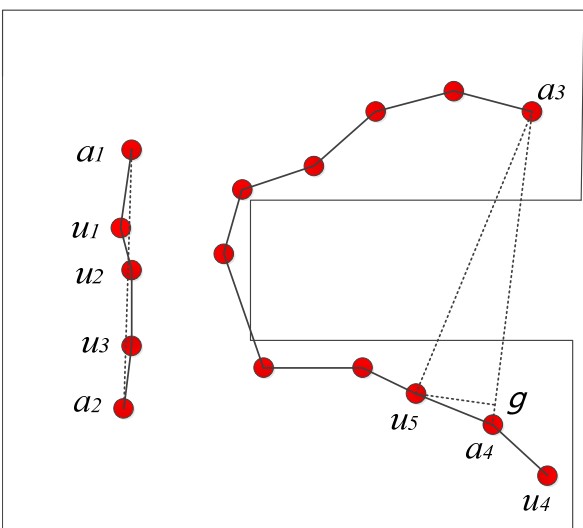

**Figure 6.** Schematic diagram of distance calculation.

### 2.3.2. Location Calculation of Unknown Nodes

$n$ is the amount of beacon nodes in the sensor network, the coordinates of the beacon nodes are: $a_1(x_1, y_1) \cdots \cdots a_i(x_i, y_i) \cdots \cdots a_n(x_n, y_n)$. The distances from $u$ to the beacon nodes are $d_{1u} \cdots \cdots d_{iu} \cdots \cdots d_{nu}$, so:

$$\begin{cases} (x_1 - x_u)^2 + (y_1 - y_u)^2 = d_{1u}^2 \\ \vdots \\ (x_i - x_u)^2 + (y_i - y_u)^2 = d_{iu}^2 \\ \vdots \\ (x_n - x_u)^2 + (y_n - y_u)^2 = d_{nu}^2 \end{cases} \tag{21}$$

$$\begin{cases} x_1^2 - x_n^2 - 2x_u(x_1 - x_n) + y_1^2 - y_n^2 - 2y_u(y_1 - y_n) = d_{1u}^2 - d_{nu}^2 \\ \vdots \\ x_i^2 - x_n^2 - 2x_u(x_i - x_n) + y_i^2 - y_n^2 - 2y_u(y_i - y_n) = d_{iu}^2 - d_{nu}^2 \\ \vdots \\ x_{n-1}^2 - x_n^2 - 2x_u(x_{n-1} - x_k) + y_{n-1}^2 - y_n^2 - 2y_u(y_{n-1} - y_n) = d_{(n-1)u}^2 - d_{nu}^2 \end{cases} \tag{22}$$

Formula (22) is expressed as a matrix: CX = D

$$
D = \begin{bmatrix}
x_1^2 - x_n^2 + y_1^2 - y_n^2 + d_{nu}^2 - d_{1u}^2 \\
\vdots \\
x_i^2 - x_n^2 + y_i^2 - y_n^2 + d_{nu}^2 - d_{iu}^2 \\
\vdots \\
x_{n-1}^2 - x_n^2 + y_{n-1}^2 - y_n^2 + d_{nu}^2 - d_{(n-1)u}^2
\end{bmatrix}
\tag{23}
$$

$$
C = \begin{bmatrix}
2(x_{(n-1)} - x_n) & 2(y_1 - y_n) \\
\vdots & \vdots \\
2(x_i - x_n) & 2(y_i - y_n) \\
\vdots & \vdots \\
2(x_{(n-1)} - x_n) & 2(y_{(n-1)} - y_n)
\end{bmatrix}
\tag{24}
$$

Using the least squares method:

$$
X = (C^T C)^{-1} C^T D
\tag{25}
$$

$$
X = \begin{bmatrix} x_u \\ y_u \end{bmatrix}
\tag{26}
$$

According to Formula (26), the location of the target node $u$ is $(x_u, y_u)$.

## 3. Improved PSO Algorithm

Figure 6 manifests that if the deployment of sensor nodes in this area is extremely sparse, although the unknown node $u_5$ is on the MSP, the node $u_5$ may also be far away from $a_3$ compared with $a_4$; in the same way, although the node $u_4$ is outside the MSP, it is also possible closer to $a_3$ than $a_4$. Therefore, in the case of extremely sparse node deployment, there will be a slight error in the calculation of dissimilar paths using Formula (16). However, constrained by the *Ochiai* coefficient and the distance between beacon nodes on the MSP, the error can be ignored. In view of this situation, this paper uses PSO to iteratively optimize the localization results of the first stage.

The PSO algorithm has the advantages of few parameters, faster convergence, no need for gradient information and so on [13]. In the algorithm, using the massless particle swarm to looking for the optimal location. The basic principle of the PSO algorithm is: If there are $m$ particles in the particle swarm, looking for the optimal solution in the space with D-dimensional, then the location of the $i$-th particle is: $X_i = (x_{i1}, x_{i2}, \cdots\cdots x_{iD})$. Each location of the particle is a potential solution, and the pros and cons of the position are measured according to the objective function. The optimal location that the particle passed is: $P_{best(i)} = (p_{i1}, p_{i2}, \cdots\cdots p_{iD})$, record the optimal coordinate currently searched in the entire particle swarm as: $G_{best} = (p_{g1}, p_{g2}, \cdots\cdots p_{gD})$. During the execution, particles find better $G_{best}$ by iteratively updating their moving speed and location. The updated iterative formulas for the speed and location of the particles are:

$$
v_{id}^{t+1} = \omega_t \cdot v_{id}^t + c_1 r_1 (p_{id}^t - x_{id}^t) + c_2 r_2 (p_{gd}^t - x_{id}^t)
\tag{27}
$$

$$
x_{id}^{t+1} = x_{id}^t + v_{id}^t
\tag{28}
$$

In the Formulas (27) and (28), $i = 1, 2, \cdots m$, $d = 1, 2, \cdots D$, $t$ is the current iteration number, $P_{best(i)}$ is the individual extreme value of the $i$-th particle, and $G_{best}$ is the current optimal solution. The random numbers are $r_1, r_2$, whose values are between in [0,1], $\omega$ is the inertia weighting factor, the acceleration factors are $c_1, c_2$, which enable particles to have the ability to learn, so as to approach the individual best points and the global best points.

The research uses PSO algorithm in the second stage to iteratively optimize the results obtained in the first stage. With the aim of declining the impact of distance error on the results and improving localization accuracy. The location of beacon nodes are: $a_1(x_1, y_1) \cdots a_i(x_i, y_i) \cdots a_n(x_n, y_n)$, According to the first-stage ranging algorithm, the distances from beacon nodes to $u$ are: $d_{1u} \cdots d_{iu} \cdots d_{nu}$. If the errors of distance estimation are: $\varepsilon_{1u} \cdots \varepsilon_{iu} \cdots \varepsilon_{nu}$, we can get the following Formula (29):

$$\begin{cases} \sqrt{(x_1 - x_u)^2 + (y_1 - y_u)^2} = d_{1u} + \varepsilon_{1u} \\ \vdots \\ \sqrt{(x_i - x_u)^2 + (y_i - y_u)^2} = d_{iu} + \varepsilon_{iu} \\ \vdots \\ \sqrt{(x_n - x_u)^2 + (y_n - y_u)^2} = d_{nu} + \varepsilon_{nu} \end{cases} \tag{29}$$

The value of $|\varepsilon_{1u}| + \cdots + |\varepsilon_{iu}| + \cdots + |\varepsilon_{nu}|$ is smaller, the result is more accurate, so the fitness function is designed as:

$$F = \sum_1^n \left| \sqrt{(x_i - x_u)^2 + (y_i - y_u)^2} - d_{iu} \right| \tag{30}$$

The specific steps of using particle swarm to optimize the coordinates of sensor nodes are as follows:

1.  Initialize particle population. The difference between the two-stage localization algorithm and other localization algorithms is that the initial position of the particle population is not completely random when locating the unknown node $u$ [14], but is limited to a specific circle. The center of the circle is the localization result $(x_u, y_u)$ of node $u$ obtained in the first stage. The radius is the maximum value $D_{DSP}$ of all non-similar path lengths of the unknown node $u$ in the ranging stage. The calculation formula of $D_{DSP}$ is:

$$D_{DSP} = max\{d_{DSP1}, d_{DSP2}, \cdots \cdots d_{DSPi}\} \ (i = 1, 2, 3, \cdots \cdots) \tag{31}$$

That is, the initial location of the particle population is randomly generated within the circle, assigning the initial speed of all particles to 0, and the maximum iteration $t_{max}$ to 100.
2.  Figure out the fitness function value of every particle base on Formula (30).
3.  Update the $P_{best(i)}$ and $G_{best}$ base on the Formulas (32) and (33):

$$P_{best(i)}^{t+1} = \begin{cases} X_i^{t+1}, F(X_i^{t+1}) \le F(P_{best(i)}^t) \\ P_{best(i)}^t, F(X_i^{t+1}) > F(P_{best(i)}^t) \end{cases} \tag{32}$$

$$G_{best}^{t+1} = \begin{cases} P_{best(i)}^{t+1}, F(P_{best(i)}^{t+1}) \le G_{best}^t) \\ G_{best}^t, F(P_{best(i)}^{t+1}) < G_{best}^t) \end{cases} \tag{33}$$

4.  According to Formulas (27), (28) and (34), the coordinates and speed of each particle are updated. The smaller the $\omega$ value, the stronger local search capability. The larger the value of $\omega$, the stronger global search capability. Because the optimization is based on the existing results, the particles need strong local search capability, and there is no requirement for the global search ability of the particles [15]. So it will be designed $\omega_t$ as a function that gradually decreases as the times of iterations increases. That is, as the times of iterations increases, the local optimization ability of

the particles is gradually increased. Lots of experiments have proved that the highest localization accuracy is obtained when $\omega_{min} = 0.25$ and $\omega_{max} = 0.75$. $\omega_t$ is the current inertia factor.

$$\omega_t = \omega_{max} - \frac{t(\omega_{max} - \omega_{min})}{t_{max}} \tag{34}$$

5.  If $t_{max} = 100$, the algorithm ends, and the $G_{best}$ is taken as the final coordinate of the target unknown node. Otherwise, perform step 2.
6.  Select the next unknown node and operate step 1.

The pseudo code of the improved PSO algorithm in the second stage is shown in Algorithm 1.

---

**Algorithm 1** The Improved PSO Algorithm

---

**1:** $D_{DSP} = max\{d_{DSP1}, d_{DSP2}, \cdots\cdots d_{DSPi}\}$ $(i = 1, 2, 3, \cdots\cdots)$
**2:** Initial population, $X_i = (x_{i1}, x_{i2}, \cdots\cdots x_{iD})$
**3:** $t_{max} = 100, v_{id}^t = 0, k = 0, t = 0$
**4:** for $k = 1, 2, \cdots m$
**5:**     for $t = 1, 2, \cdots t_{max}$
**6:**         figure out the $F$ value of every particle
**7:**         if $F(X_i^{t+1}) \leq F(P_{best(i)}^t)$
**8:**             $P_{best(i)}^{t+1} = X_i^{t+1}$
**9:**         else $P_{best(i)}^{t+1} = P_{best(i)}^t$
**10:**        endif
**11:**        if $F(P_{best(i)}^{t+1}) \leq G_{best}^t)$
**12:**            $G_{best}^{t+1} = P_{best(i)}^{t+1}$
**13:**        else $G_{best}^{t+1} = G_{best}^t$
**14:**        endif
**15:**        $\omega_t = \omega_{max} - t(\omega_{max} - \omega_{min})/t_{max}$
**16:**        $v_{id}^{t+1} = \omega_t \cdot v_{id}^t + c_1 r_1 (p_{id}^t - x_{id}^t) + c_2 r_2 (p_{gd}^t - x_{id}^t)$
**17:**        $x_{id}^{t+1} = x_{id}^t + v_{id}^t$
**18:**        output $x_{id}^{t_{max}}$
**19:**    endfor
**20:** endfor

---

## 4. Experiment Results and Analysis

Experiments were carried out under the Windows 10 computer operating system, and matlab2016a was used for simulation experiment.

$$AverageError = \sum \frac{\sum_u \sqrt{(x - x_u)^2 + (y - y_u)^2}}{R(NAmount - BAmount)} \times \frac{1}{T} \tag{35}$$

In Formula (35), $T$ represents the times of experiments, $(x, y)$ is the actual coordinate of node $u$, the estimated coordinate is $(x_u, y_u)$, communication radius is $R$, the total amount of sensor nodes is *NAmount*, the amount of beacon nodes is *BAmount*.

A concave boundary was randomly generated within the square communication region which with the area of 40,000 square meters, so that the communication region became a concave region, then, 100 nodes were randomly deployed in the communication area to simulate various sensor networks by changing communication radius and proportion of beacon nodes in the WSN. Then, we studied the influence of these parameter changes on the results and algorithm execution time.

### 4.1. The Impact of Beacon Node Ratio on Localization Results and Execution Time of Algorithm

#### 4.1.1. Relationship between Beacon Node Ratio and Localization Results

We can see from Figure 7 that the localization accuracy of FABL algorithm, CHP algorithm and REP algorithm fluctuated greatly with the change of beacon node ratio, and when the ratio is less than 15%, the error of these three algorithms was more than 20%. This is because when the beacon nodes were relatively sparse, the ranging method based on the hop count multiplied by the hop distance caused larger localization errors due to the accumulation of hop distance errors [16]. This study invented a method that based on the intersection ratio to calculate the distance, which did not depend on the coordinate of the target beacon nodes, and discarded the traditional ranging idea of multiplying the number of hops by the hop distance, effectively avoiding the hop error and improving the accuracy. In Figure 7, the error of two-stage PSO algorithm proposed in this paper was always within 10%. Compared with the others three methods, it was less affected by the proportion of beacon nodes and had higher localization accuracy.

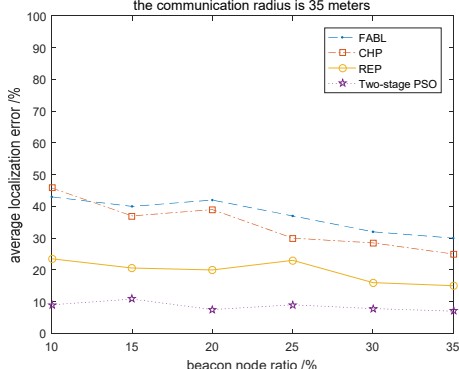

**Figure 7.** Relationship between beacon node ratio and localization results.

#### 4.1.2. Relationship between Beacon Node Ratio and Execution Time of Algorithm

In Figure 8, the REP algorithm took more than 110 s to execute; the higher the beacon node ratio, the longer the execution time required for the algorithm. The execution time of CHP algorithm was between 70 s and 80 s, and the execution time of FABL algorithm and two-stage PSO algorithm was about 20 s. The REP algorithm and the CHP algorithm needed to construct a virtual circle and divide the communication area, so more calculation time was required. The two-stage PSO algorithm did not need to split the path, divide the communication area operations, just by judging whether the path between the nodes was affected by the concave boundary, and could complete the ranging. The PSO was used in this algorithm, but optimization on the basis of the existing results required very few iterations and took a short time.

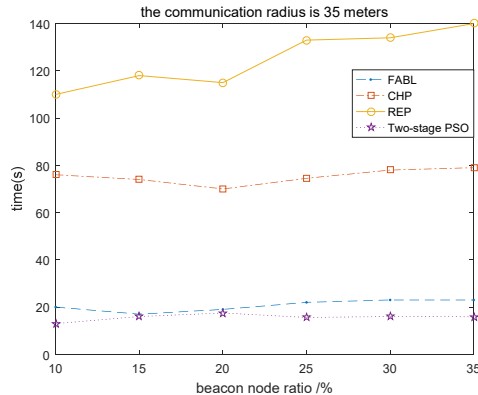

**Figure 8.** Relationship between beacon node ratio and execution time of algorithm.

### 4.2. The Impact of Node Communication Radius on Localization Results and Execution Time of Algorithm

4.2.1. Relationship between Node Communication Radius and Localization Results

Figure 9 indicates that when radius was less than 20 m, the average error of FABL and CHP were both above 45%, the error of the REP also reached 29%. This is because when the communication radius was small, the hop count on the shortest path suddenly increased, which resulted in a large accumulation of hop distance error during the ranging process and greatly reduced the accuracy of localization [17]. Although the localization accuracy of these three algorithms was improved with the increase of radius, the coordinate error was still large, and an excessive communication radius greatly increased the cost of the network. In Figure 9, the average error of the two-stage PSO algorithm was always within 10%, which had the higher localization accuracy. The algorithm was unacted on the radius, and was more suitable for the localization of complex and diverse concave regions.

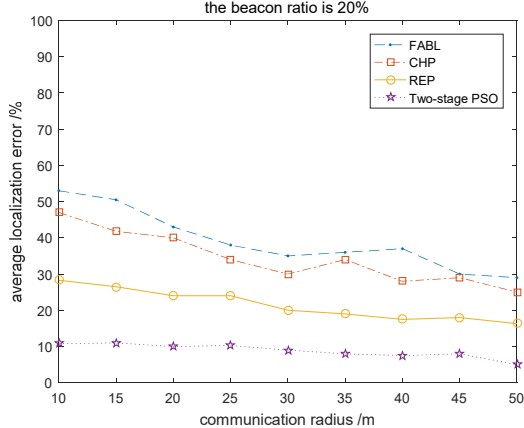

**Figure 9.** Relationship between node communication radius and localization results.

4.2.2. Relationship between Communication Radius and Execution Time of Algorithm

In Figure 10, the REP algorithm and the CHP algorithm in the four algorithms took a long time. The execution time of the REP was about 120 s, and the execution time of the CHP was about 80 s. The execution time of these two algorithms was mainly consumed in the process of path segmentation and region partitioning, so it was not affected by the radius. The execution time of FABL algorithm and two-stage PSO algorithm was relatively short. As the radius changed, the execution time of these two algorithms changed slightly. This was because as the radius increased, the amount of hops between nodes gradually decreased, reducing the ranging time. On the whole, compared with other algorithms, the execution time of the two-stage PSO was shorter, which could extend the service life of the node.

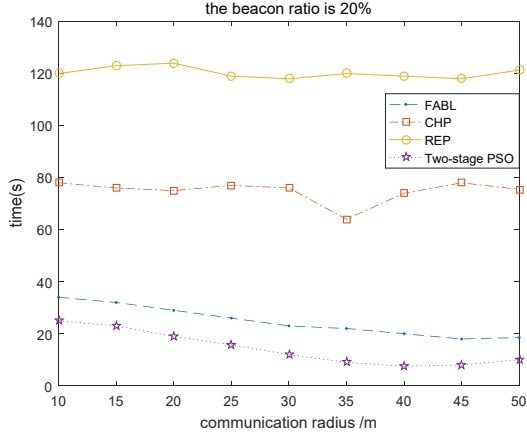

**Figure 10.** Relationship between communication radius and execution time of algorithm.

## 5. Conclusions

For the problem of the nodes localization algorithm in the concave area, this paper studied the deficiencies of various concave area localization algorithms based on beacon node selection, communication area division, shortest distance correction, etc. Considering the shortcomings of intelligent optimization algorithm in WSN location, the energy consumption is too large and the result is not qualified. In this paper, a two-stage PSO node localization algorithm which suitable for concave areas was proposed. The algorithm creatively uses the ranging method based on the idea of similar path and intersection ratio in the node location of concave area, and combines it with the PSO algorithm. It not only effectively improves the accuracy of the distance, but also solves the problem that the amount of iterations of the intelligent optimization methods in the WSN localization application is too large. Through a lots of experiments, it is proved that the localization error of this algorithm is always within 10% and the execution time is maintained at about 20 s when the communication radius and beacon node ratio is changing. So, the algorithm has good localization accuracy and good stability, which can greatly reduce the computational energy consumption. In the next step of research, we will continue to study the other algorithms that the localization of wireless sensor nodes, as well as the impact of beacon node coordinates on the localization results.

**Author Contributions:** Conceptualization, Y.M. and Q.Z. (Qianying Zhi); formal analysis, Q.Z. (Qiuwen Zhang); funding acquisition, Y.M.; investigation, Q.Z. (Qianying Zhi) and Q.Z. (Qiuwen Zhang); methodology, Q.Z. (Qianying Zhi); project administration, Y.M.; software, Ni Yao; supervision, Y.M.; visualization, N.Y.; writing—original draft, Q.Z. (Qianying Zhi); writing—review and editing, Q.Z. (Qianying Zhi). All authors have read and agreed to the published version of the manuscript.

**Funding:** This research was funded by National Natural Science Foundation of China, grant number 61501405, and 61771432. Science and Technology Planning Program of Henan Province, grant number 202102210398.

**Conflicts of Interest:** The authors declare no conflict of interest.

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
