# Peer review of "A Two-Stage Particle Swarm Optimization Algorithm for Wireless Sensor Nodes Localization in Concave Regions"

_information, doi:10.3390/info11100488_

Round 1

Reviewer 1 Report

The idea of initializing the particles by a Euclidean distance inside a circle is adequate and the results claim its efficiency. Despite the good results of the localization proposal and aligned to the main requirements of a WSN, which is caring about the energy consumption, It is important to show the results of radio activity during the bootstrap phase. How much energy is spent to set the locations of the nodes? What is the influence of W on radio activity to share the information?

Reviewer 2 Report

I could appreciate the logic of the manuscript and made myself convinced that the paper could be interesting to the relevant scientific community.

See details in the attachment.

Reviewer 3 Report

As the authors stated in the abstract, node localization is an important problem in WSN.  There are not many challenges in convex space. However, concave localization throws more challenges. Algorithmic complexity is one of the key issues in designing efficient solution. In this paper, the authors have proposed a two stage Particle Swarm optimization algorithm for WSN in a concave region.

The paper is easy to read, well-written and contain publishable results. This is the strength. However, I have the following suggestions for improving the overall presentation of this version:

  1. Formulate your algorithm in Pseudo code; so, it may be easy for readers. Your calculation and individual process can be regarded as a “module” in your algorithm.
  2. Once the algorithm is established, you need to establish “proof of correctness” and complexity analysis (as mentioned in L11-12).

The above two improvements may substantially improve the overall presentation of this paper.

Reviewer 4 Report

The paper is well written and was based on a detailed mathematical judgment.

Abstract

It is a good practice to mention evaluation parameters used in the research and the significant results in percentage achieved with the proposed algorithm.

1. Introduction

1.1. Research Significance

Include in the paper the references that you based your analysis of the range-free and range-based localization algorithms on.

I can’t see reference [1] in the text.

1.2. Research Status

Line 101- specify which one is the “first kind of localization algorithm”? Does this sentence belong to the previous paragraph? It is not clear.

Same for “second kind of localization”.

2. Initial Localization of Unknown Nodes

2.1. Calculate the Euclidean Distance of One Hop Based on (small letter) the Intersection Ratio

Line 151- where is “communication circle b”?

2.2. Judge Whether the Multi-hop Shortest Path Between Beacon Nodes is Affected by Concave Boundary

This section should include an introductory paragraph for the coming sections and the reasoning behind them.

2.2.1. Judgment Method

Line 171- it is better to mention the section instead of saying “will be given below”.

The authors should clearly mention if this part of research is based on previous work and cite it.

4. Experiment Results and Analysis

4.1. The Impact of Beacon Node Ratio on Localization Results and Execution Time of Algorithm

This section should include an introductory paragraph for the algorithms simulated.

5. Conclusions

Results with percentages should be mentioned with more details according to the evaluation parameters used.

References

Correct all your citations, there is missing information. For example, reference 11 and 13 to 17 are without the journals name.

Also, the name of the journal or conference should be in italic. For example, check reference 4, 9 and 10.
